# Sutureless Amniotic Membrane Transplantation in Inflammatory Corneal Perforations

Alessandro Meduri, Antonio Valastro, Leandro Inferrera , Giovanni William Oliverio *, Ivan Ninotta, Umberto Camellin, Maura Mancini, Anna Maria Roszkowska and Pasquale Aragona

Biomedical Science Department, Institute of Ophthalmology, University of Messina, Via Consolare Valeria 1, 98125 Messina, Italy; alessandro.meduri@unime.it (A.M.); antonio.valastro2@gmail.com (A.V.); inferreraleandro@gmail.com (L.I.); ivan.ninotta@gmail.com (I.N.); umberto.camellin.uc@gmail.com (U.C.); maura.mancini94@gmail.com (M.M.); anna.roszkowska@unime.it (A.M.R.); pasquale.aragona@unime.it (P.A.)
* Correspondence: gioliverio@unime.it; Tel.: +39-0902212279

**Abstract: Introduction:** The aim of this study was to evaluate the efficacy of sutureless amniotic membrane transplantation (SAMT) in patients with corneal perforation secondary to ocular surface inflammatory diseases. **Methods:** Twelve eyes of eleven patients with corneal perforation associated with Sjögren's syndrome and ocular cicatricial pemphigoid were included. Surgical procedure consisted in the application of two layers of amniotic membrane covering the corneal perforation and a therapeutic contact lens placed above the two layers, followed by sutureless tarsorrhaphy with Steri-strip applied for 15 days to immobilize the eyelids. All patients underwent slit-lamp examination, fluorescein corneal staining, and anterior segment optical coherence tomography (AS-OCT), preoperatively and post-surgery, weekly for one month, and then every two weeks for three months. **Results:** All 12 eyes had complete resolution of corneal perforation. Pachymetry measurements improved in all eyes, and complete healing of the corneal ulcer was obtained following treatment and maintained up to 3 months follow-up in all patients. **Conclusions:** SAMT was proven to be a safe and effective option for the management of inflammatory corneal perforation. The procedure is non-traumatic and easy to perform since the surgical technique is sutureless and does not require topical therapy because it takes advantage of the intrinsic properties of the membrane itself.

**Keywords:** amniotic membrane; cornea; transplantation; corneal perforation; sutureless; anterior segment optical coherence tomography (AS-OCT)

## 1. Introduction

Amniotic membrane (AM), or amnion, is the deepest of the three layers that constitute fetal membranes. It is a semi-transparent membrane that consists of a single layer of amnion epithelial cells fixed to a large basement membrane, and an avascular stromal matrix [1]. It has been demonstrated that the human amniotic membrane contains collagen types IV and VII, such as with the Bowman membrane cornea, in addition to integrin, fibronectin, and laminin [2,3]. It was employed for the first time for therapeutic use in 1910 by Davis for skin transplantations [4], while its first application in ophthalmology was described by De Rotth, who used the fetal membrane (both amniotic membrane and chorion) to treat epithelial conjunctival defects in patients with symblepharon [5]. Several studies have demonstrated the different properties of amniotic membrane, such as pain reduction, anti-inflammatory activity, and antiadhesive and antiangiogenic effects [6–11]. In the last two decades, the amniotic membrane has progressively been used to treat different ophthalmic conditions, such as chemical burns, persistent corneal epithelial defects, reconstruction of ocular surfaces, ocular pemphigoid, Stevens-Johnson syndrome, and bullous keratopathy [1,12–15]. Depending on the underlying pathology, three main surgical techniques can be used.

The first one is the inlay or graft technique and is used for stromal defects. The AM is put with the epithelial side up. The graft is fixed with 10-0 nylon sutures at the periphery of the corneal defect. The epithelium is expected to grow above the AM [16]. The second one is the overlay or patch technique, used for non-healing epithelial defects with little or no stromal involvement (i.e., chemical burns or recurrent corneal erosions). The AM can be used with either the epithelial side or the stromal side up and is sutured beyond the epithelial defects and the central stromal defect [16]. The epithelium is expected to grow under the AM. The third one is the sandwich technique, used for deep stromal defects with large non-healing epithelial defects [16]. With this technique, two or more layers of AM are used; the inner one acting as a graft and the outer one acting as a patch. The epithelium is expected to grow between the two layers [16]. What all these techniques have in common is that they require the use of sutures, which can cause several complications. It is for this reason that in recent years surgical techniques that do not involve the use of sutures have been developed, such as cyanoacrylate and fibrin glue, which are the most widely used.

In this case series, we used a sutureless amniotic membrane transplantation (SAMT) in patients with corneal perforation secondary to ocular surface inflammatory diseases. The technique consists in applying a double layer of AM and a therapeutic contact lens on the corneal surface of the patient, followed by the eyelid closure with Steri-strip™. It can be done with relative ease also by general ophthalmologists who are not necessarily corneal specialists.

## 2. Materials and Methods

In this retrospective study, 12 eyes of 11 patients subjected to AM graft surgery between July 2020 and September 2021 were studied. Inclusion criteria were severe corneal thinning, descemetoceles, or perforation (with positive Seidel test). Exclusion criteria were corneal infectious diseases and traumatic perforation.

The study included 12 eyes of 11 patients (10 females and 1 male, 5 with Sjögren's syndrome and 6 with ocular cicatricial pemphigoid).

All patients were non-responsive to medical treatment and suffered from a pathology that led to corneal perforation that required immediate intervention.

All surgeries were performed by the same surgeon (AM) at the ophthalmic department of the University of Messina, Italy. The study was conducted with respect of tenets of Declaration of Helsinki and obtained approval of the Ethical Committee of the University Hospital of Messina.

The 12 eyes were evaluated the day after surgery, weekly for one month and then every two weeks for three months.

Corrected distance visual acuity (CDVA) using LogMAR chart, slit-lamp biomicroscopy examination (Topcon SL-D701, Topcon, Essebaan, LJ Capelle a/d IJssel, the Netherlands), fluorescein corneal staining, and anterior segment optical coherence tomography (AS-OCT) (iVue, Optovue, Bayview Drive Fremont, CA, USA) were preoperatively evaluated. Corneal pachymetric values were measured manually, using the caliper tool software integrated with AS-OCT, in the thinnest part of the ulcer. These values were measured by two different operators, and the difference between the two measurements was not statistically significant. The mean value of the two measurements was used as the corneal pachymetric value.

### 2.1. Surgical Technique

Disinfection of the eyelid skin was carried out with 5% iodopovidone. The eye was draped, and a lid speculum was placed. Oxybuprocaine hydrochloride (Novesina™ 0.4%, Laboratoires Thea) was applied as preoperative anesthesia. Povidone-iodine 5% was applied for three minutes to allow disinfection of the ocular surface. The membrane was initially cropped with Westcott scissors to create a first layer, which acts as a graft, with a diameter of 4 mm greater than the perforation (Figure 1a). This graft was applied above the corneal perforation (Figure 1b). A second layer of AM, which acts as a patch and is

larger than the previous one, was then placed on top of that (Figure 1c). Each AM layer was placed epithelial side up. A contact lens, with a diameter of 12 mm, was placed above these 2 layers (Figure 1d,e). Finally, a 3M™ Steri-strip™ R-100 × 12 mm was positioned to occlude the eyelids (Figure 1f). No postoperative eye drops were prescribed. The eyelid closure and the therapeutic contact lens (Pure Vision 2™ Bausch + Lomb, Bridgewater, NJ, USA) were maintained up to two weeks. In addition, patients were given oral doxycycline 100 mg bid for 15 days.

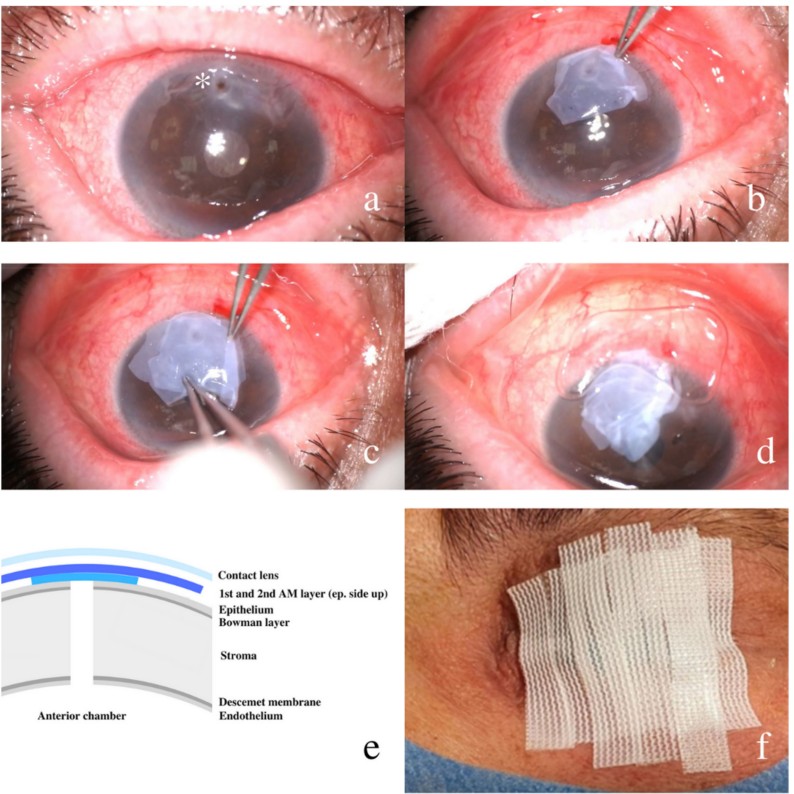

**Figure 1.** Corneal perforation (*) (**a**). First layer of amniotic membrane (AM) covering the corneal perforation (**b**). Second layer of AM placed on top of the first AM layer (**c**). Contact lens (12 mm diameter) placed above the 2 AM layers (**d**). Schematic representation of result of the surgical technique showing the 2 layers of AM covering the corneal perforation. The contact lens is shown on top of them (**e**). 3M™ Steri-strip™ R-100 × 12 mm used to close the eyelids (**f**).

*2.2. Statistical Analysis*

The Shapiro-Wilk test for normality distribution data was used. For non-parametric data, the Friedman test and Wilcoxon rank-sum post hoc test were used. Results with a P-value of less than 0.05 were considered statistically significant using Bonferroni correction (P with Bonferroni correction = 0.0083). In order to account for the mixed use of both eyes of the same patient versus only one eye per patients for some patients [17], the statistical analyses were run using the number of patients (and not the number of eyes). To account for enantiomorphism between eyes [18], the parameters for the controlateral eye was mirrored. With the aim of canceling bias related to position of the corneal ulcer and to pachymetry of each individual patient, we decided to study the pachymetric variation during follow-up. Statistical analysis was performed using SPSS (Statistical Package for Social Science, IBM Corp., Armonk, NY, USA).

**3. Results**

Twelve eyes of eleven patients (10 females, 1 male) treated for Sjögren's syndrome and ocular cicatricial pemphigoid were enrolled (1 patient bilateral and 11 patients unilateral).

CDVA changed from 1.82 ± 0.64 logMAR preoperatively to 1.68 ± 0.58 logMAR 3 months at postoperatively. Slit-lamp biomicroscopy examination and fluorescein corneal staining confirmed the corneal perforation healing. The preoperative mean corneal pachymetry (μm) and postoperative at 2, 4, 8, and 12 weeks are shown at Table 1.

**Table 1.** Pre-operative and post-operative corneal pachymetry calculated at the thinnest point of the ulcer with anterior segment optical coherence tomography (AS-OCT).

| | Pre-Operative | 2 Weeks Postop | 4 Weeks Postop | 8 Weeks Postop | 12 Weeks Postop | Δ 2 Weeks-Preop | Δ 4 Weeks–2 Weeks | Δ 8 Weeks–4 Weeks | Δ 12 Weeks–8 Weeks |
|---|---|---|---|---|---|---|---|---|---|
| | (μm) | (μm) | (μm) | (μm) | (μm) | (μm) | (μm) | (μm) | (μm) |
| Mean | 0.00 | 318.33 | 345.92 | 349.50 | 352.67 | 318.33 | 27.58 | 3.58 | 3.17 |
| Sd | 0.00 | 85.48 | 84.78 | 84.02 | 84.64 | 85.48 | 26.85 | 2.02 | 3.51 |
| Median | 0.00 | 317.50 | 369.50 | 371.00 | 374.00 | 317.50 | 21.00 | 3.50 | 2.00 |
| Q1 | 0.00 | 262.75 | 304.50 | 311.25 | 320.00 | 262.75 | 10.00 | 3.00 | 1.50 |
| Q3 | 0.00 | 369.75 | 386.50 | 389.50 | 391.75 | 369.75 | 29.75 | 4.25 | 3.50 |
| Variance | 0.00 | 7306.97 | 7187.72 | 7060.09 | 7164.24 | 7306.97 | 720.99 | 4.08 | 12.33 |
| Min | 0.00 | 195.00 | 205.00 | 210.00 | 212.00 | 195.00 | 7.00 | 0.00 | 0.00 |
| Max | 0.00 | 475.00 | 497.00 | 501.00 | 510.00 | 475.00 | 105.00 | 8.00 | 11.00 |

AM engraftment resulted in being satisfactory in all 12 eyes with complete resolution of the corneal perforation and re-epithelialization with anterior chamber formation. No complications developed, neither during the SAMT surgical procedure nor three months post-op in all the 12 eyes.

The mean postoperative pachymetry changing is shown in Table 1, and statistical analysis is shown in Table 2 and Figure 2.

**Table 2.** Corneal pachymetric variation after treatment and during follow-up.

| | Δ 4 Weeks–2 Weeks | Δ 8 Weeks–4 Weeks | Δ 12 Weeks–8 Weeks |
|---|---|---|---|
| **Δ 2Weeks-Preop** | Z = −3.059, $p < 0.0083$ | Z = −3.059, $p < 0.0083$ | Z = −3.059, $p < 0.0083$ |
| **Δ 4Weeks–2 Weeks** | | Z = −3.059, $p < 0.0083$ | Z = −3.064, $p < 0.0083$ |
| **Δ 8Weeks–4 Weeks** | | | Z = −3.064, $p < 0.0083$ |

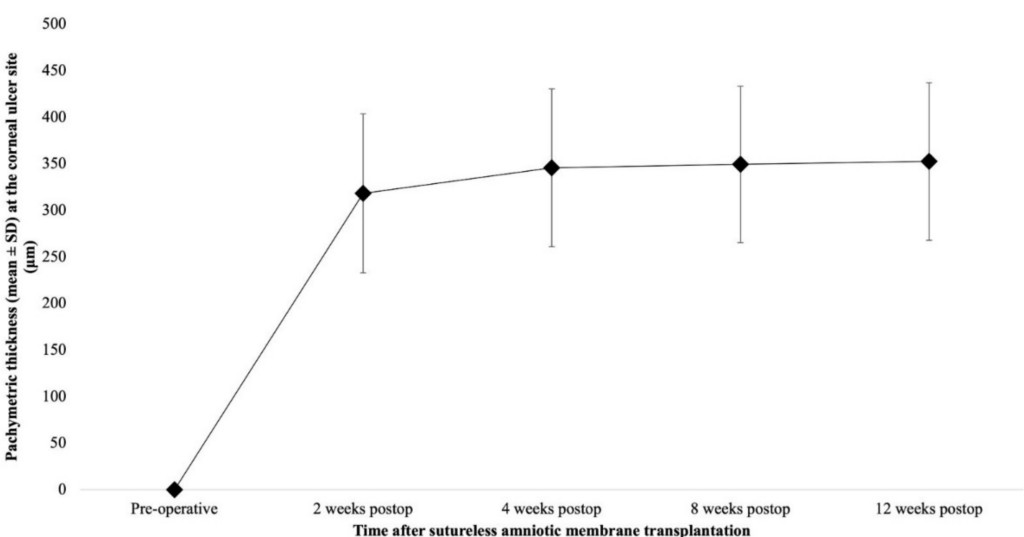

**Figure 2.** Graph shows the mean pachymetric variation (μm) and its standard deviation (SD) at the level of treated corneal ulcer.

All 12 eyes showed indeed an excellent tectonic result as shown by AS-OCT (Figure 3).

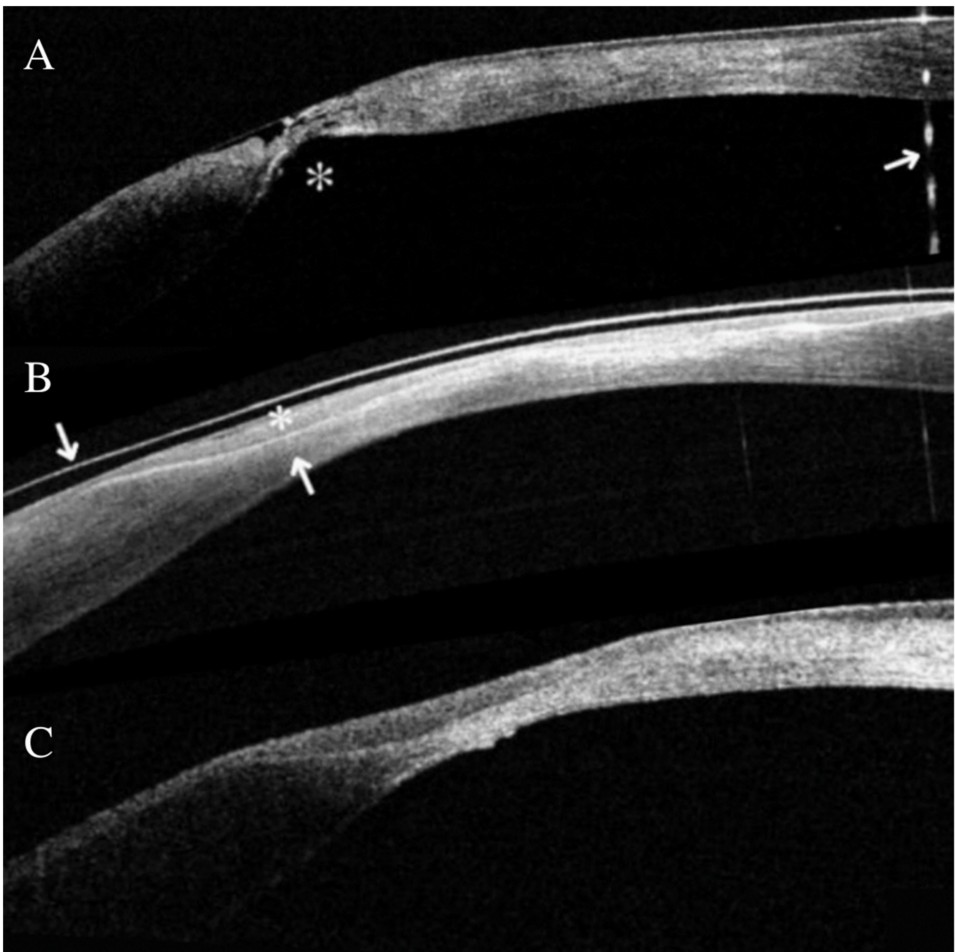

**Figure 3.** Patient number 3, right eye. (**A**) Anterior segment optical coherence tomography (AS-OCT) showing corneal thinning with residual stromal tissue (asterisk), Seidel positive. A reflex saturation beam artifact is visible as a dashed line (arrow). (**B**) The postoperative outcome at week 1. The therapeutic lens is seen above the corneal epithelium as a hyper-reflective band (arrow). The epithelium regrowth is indicated by a small asterisk and delimited by the Bowman's membrane, visible as a hyper-reflective band (arrow). (**C**) The post-op outcome at 1 month shows a similar corneal appearance to week 1.

## 4. Discussion

Corneal stromal wound healing is a very complex and orderly process with keratocyte death and repopulation, sequential transformation of keratocytes into fibroblasts and myofibroblasts, infiltration of limbal and circulating immune cells, and remodeling of the corneal extracellular matrix (ECM) structure [19–22].

Several properties explain the beneficial effects of AM transplantation for ocular surface reconstruction. Foremost, the AM has a mechanical function resulting in the growth of epithelial cells on the ocular surface. Furthermore, it stimulates epithelialization through the production of the epithelial growth factor (EGF), hepatocyte growth factor (HGF), and keratinocyte growth factor (KGF). Ultimately, the AM reinforces the adhesion of basal epithelial cells and prevents epithelial cell apoptosis. The inhibition of the proliferation of corneal, limbal, and conjunctival fibroblasts results from the suppression of TGFb signaling along with the reduced expression of TGF b-1, b-2, and b-3 isoforms and TGF-beta receptor II, which entails the anti-fibrotic effect of the AM [23].

The anti-inflammatory effect results from the inhibition of the expression of pro-inflammatory cytokines (IL-1a, IL-2, IL-8, IL-10, IFN-c, bFGF, TNFb, and PDGF) from the damaged ocular surface [24]. Additionally, it has been demonstrated that the AM stroma

acts as a trap for the inflammatory cells that undergo apoptosis [24]. Moreover, AM stromal and epithelial cells express FAS receptor, driving the fibroblast and inflammatory cells to apoptosis. The lack of immunogenicity avoids the need of any immunosuppressive treatment, enhancing the benefit of AMT. Finally, AM anti-angiogenic and anti-microbial properties have been demonstrated [24]. Current AMT techniques involve suturing the graft or patch of AM over the ocular surface. Nevertheless, the corneal sutures can not only cause postoperative pain and discomfort to patients [25] but also lead to complications such as additional trauma, suture abscesses [26,27], infections [28,29], granuloma formation [30], and tissue necrosis [31]. Furthermore, the inflammatory effect of corneal suture [32] can lead to an unsuccessful repairing attempt, especially if the corneal damage is due to an inflammatory disease. For this reason, sutureless techniques have been developed, including the use of cyanoacrylate glue. Although proven effective, this type of glue can cause complications, such as chronic inflammation and delay of wound healing due to its toxicity [33–35]. Fibrin glue is a preferred option to fix the AM to the ocular surface. The major drawback to its use is the risk of transmitted disease from blood donors [35,36]. Other SAMT-reported procedures include mounting the amniotic membrane on an ocular conformer [37] or directly on a contact lens [38]. Yi et al. managed to prepare a contact lens-shaped AM using glutaraldehyde and dialdehyde starch as a cross-linking agent [39]. The literature reports the use of a self-retaining cryopreserved amniotic membrane (PROKERA® Slim, Bio-Tissue, Miami, FL, USA) for the treatment of dry eye disease [40], corneal ulcers [41], Stevens-Johnson syndrome/toxic epidermal necrolysis [42], and Sjögren's syndrome [43]. Some authors reported the use of a symblepharon ring in order to fix the AM to the ocular surface in patients with chemical injuries or toxic epidermal necrosis/Stevens–Johnsons syndrome [44,45]. Kotomin et al., described the "AmnioClip system" for the mounting of AM between two rings for application to a diseased ocular surface without surgical intervention [46]. In our cases, we took advantage of the beneficial effects of AMT by avoiding the negative pro-inflammatory effects related to the use of sutures. In fact, they could worsen the patient's ocular surface inflammatory disease and lead to unsuccessful treatment. Moreover, our procedure is easily performed, requiring a contact lens and a bandage for the eyelid. The technique can be done with relative ease also by general ophthalmologists who are not necessarily corneal specialists. The procedure consists in cutting two layers of AM—the small one is used as a graft and the larger one acts as a patch. The grafts are not fixed to the ocular surface. A contact lens is immediately applied, and the lids closed with the use of 3M™ Steri-strip™, thus preventing the contact lens to move and the grafts to displace. We chose to utilize the tape bandage rather than the traditional suture tarsorrhaphy to avoid local complications such as trichiasis, adhesion between upper and lower lids following tarsorrhaphy lysis, premature opening of the temporary tarsorrhaphy, pyogenic granuloma, and keloid formation of the eyelid. The lid closure is not only functional in maintaining the contact lens and AM in place, but also accelerates the healing of the corneal surface [47] and facilitates the AM healing properties, creating a microenvironment without external influences. Another advantage of this technique is that patients do not need to use topic medications, making the outcome independent from the patients' therapy compliance. Our cases showed excellent anatomic results that led to the healing of the corneal perforation and to the restoration of the anterior chamber from the first postoperative visit (2-week post-op). Furthermore, there was no dislocation in all the 12 eyes studied and no corneal infections developed during the 3 months follow-up.

## 5. Conclusions

The purpose of this study is to describe an alternative SAMT technique to treat corneal perforation due to ocular surface inflammatory diseases. SAMT was shown to be effective for a rapid and complete resolution of corneal perforation without corneal infections and was sustained up to three months of follow up in 100% cases. This technique has the advantage of reducing the pro-inflammatory effects related to the sutures and to longer surgical procedures. Another advantage is the ease of the procedure and the

little discomfort for the patients who are relieved from the use of any topical treatment. Furthermore, SAMT does not require any topical therapy since eyelids are closed by tarsorrhaphy and therefore only the intrinsic properties of the amniotic membrane are exploited. The limited number of patients treated and the brief follow-up does not allow for the expression of irrefutable conclusions. Larger studies are needed to confirm the anti-inflammatory benefits of this technique in patients with ocular surface inflammatory disease. Future studies may also evaluate the possible role of SAMT in preventing corneal perforation in patients whose inflammatory ocular surface disease is no longer controlled by topical and systemic medications.

**Author Contributions:** A.M.: conception of the work, writing of the manuscript, interpretation of data, critical revision, and writing of the manuscript. L.I. and G.W.O.: writing of the manuscript, analysis of data. A.V., I.N., U.C. and M.M.: acquisition of data, writing of the manuscript. A.M.R.: supervision of the manuscript. P.A.: supervision of the manuscript, final review, and approval. All named authors meet the International Committee of Medical Journal Editors (ICMJE) criteria for authorship for this article, take responsibility for the integrity of the work. All authors have read and agreed to the published version of the manuscript.

**Funding:** This research received no external funding.

**Institutional Review Board Statement:** The study was conducted in accordance with the Declaration of Helsinki, and approved by the Institutional Review Board (or Ethics Committee) of University of Messina (protocol code 86/19 of 2 October 2019).

**Informed Consent Statement:** Written informed consent was obtained from participants (or their parent/legal guardian/next of kin) to participate in the study.

**Acknowledgments:** The authors did not receive any financial support from any public or private sources. The authors have no financial or proprietary interest in a product, method, or material described.

**Conflicts of Interest:** The authors declare no conflict of interest.

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
