# Peer review of "Sutureless Amniotic Membrane Transplantation in Inflammatory Corneal Perforations"

_applsci, doi:10.3390/app12083924_

Round 1
Reviewer 1 Report
Oliverio's group developed the SAMT procedure for the treatment of inflammatory corneal perforation. They showed that SAMT is effective, nontraumatic, and performs easily. But there is some shortage for the paper presenting and data processing and need to be improved before accepted to publish.
- In the Introduction, The authors should discuss the treatment methods for corneal perforation in clinical and compare the advantages and disadvantages of SAMT and other methods.
- Table 1 data, these data can make a figure to clearly present and put the numbers into supplemental data.
- Line 192, "Figure 2A:" change to "Figure 2: A", there is no "A, B, C" in the figure and make it more clear. "Fig. 2B:" change to "B:". Line 198: "C: The post-op outcome at 1month shows .....".
Author Response
- In the Introduction, The authors should discuss the treatment methods for corneal perforation in clinical and compare the advantages and disadvantages of SAMT and other methods.
Thank you for these constructive advices. We have included in the introduction the state of art about the surgical techniques of amniotic membrane transplantation, describing the advantages of our surgical approach, as suggested. We have highlighted the main objective of SAMT, that allow to treat corneal perforation in patients with aggressive ocular surface inflammatory diseases, reducing the surgical stress and the consequent fibrosis on the cornea.
- Table 1 data, these data can make a figure to clearly present and put the numbers into supplemental data.
Thank you for your advice. We created a figure which represents in a schematic way all the data expressed in the table
- Line 192, "Figure 2A:" change to "Figure 2: A", there is no "A, B, C" in the figure and make it more clear. "Fig. 2B:" change to "B:". Line 198: "C: The post-op outcome at 1month shows .....".
We have changed the tables and figure according with your advices.
Reviewer 2 Report
In the presented work, the authors aim to evaluate the efficacy of sutureless amniotic membrane transplantation (SAMT) in patients with corneal perforation secondary to ocular surface inflammatory diseases. It is a relevant topic to the field with promising results for clinical application. The manuscript is well structured and clear to the reader.
I consider that some minor issues should be addressed before publication.
Further specific points are described below.
1 – The number of subjects and eyes studied should be more clear stated.
Line 77 the authors stated 11 women
Line 85 the authors stated 12 eyes
Line 137 the authors stated 12 eyes (10 female, 1 male)
Please clarify.
2 – Line 87 the authors described that it was evaluated CDVA, slit-lamp biomicroscopy examination, fluorescein corneal staining, and AS-OC. However, the results for the slit-lamp biomicroscopy examination and fluorescein corneal staining was not presented.
Please include this results in the manuscript.
3 – Line 91: The authors stated that “Corneal pachymetric values were measured manually in the thinnest part of the ulcer”. Please provide more details about this procedure. The authors could use the AS-OCT image (Figure 2) to indicate it.
4 – Line 191: Figure 2. Please label the A, B and C in the figure. Indicate the location of pachymetry measurement.
5 – Line 140: What this circle symbol represents? +/-?
6 – Please correct some format issues, lines 126, 128, 130, 132, 211, 213, 215 e.g.
Author Response
1 – The number of subjects and eyes studied should be more clear stated.
Line 77 the authors stated 11 women
Line 85 the authors stated 12 eyes
Line 137 the authors stated 12 eyes (10 female, 1 male)
Please clarify.
Thank you for these comments, we apologize for these errors. We have corrected these mistakes, reporting the correct number of subjects enrolled in the study.
2 – Line 87 the authors described that it was evaluated CDVA, slit-lamp biomicroscopy examination, fluorescein corneal staining, and AS-OC. However, the results for the slit-lamp biomicroscopy examination and fluorescein corneal staining was not presented.
Please include this results in the manuscript.
We thank you for these constructive comments, we have added more findings in results section, as suggested. The clinical evaluation of these patients included visual acuity evaluation, slit lamp and fluorescein staining to evaluate the corneal perforation, and corneal pachymetry that allow us to quantify the thickness of the cornea in the perforated area. Although these assessments where performed, CDVA did not changed after treatment as the severe corneal alteration impacted negatively on visual acuity; slit lamp and fluorescein staining confirmed in all patients the healing of the corneal perforation, but this finding is more evident considering the AS-OCT findings.
3 – Line 91: The authors stated that “Corneal pachymetric values were measured manually in the thinnest part of the ulcer”. Please provide more details about this procedure. The authors could use the AS-OCT image (Figure 2) to indicate it.
Thank you for this comment that allow us to better explain a crucial concept for methodology. The corneal thinnest point in the area of perforation was evaluated using a calliper, a tool provided by the software of the iVue OCT. This is the best method to evaluate the thickness of a perforated cornea. We have improved the description of AS-OCT.
4 – Line 191: Figure 2. Please label the A, B and C in the figure. Indicate the location of pachymetry measurement.
We have changed the figure according to your advices.
5 – Line 140: What this circle symbol represents? +/-?
6 – Please correct some format issues, lines 126, 128, 130, 132, 211, 213, 215 e.g.
We have corrected these issues.